# Ex Situ and In Situ Artificial Thermo-Aging Study of the Natural Degradation of *Bombyx mori* Silk Fibroin

**Monika A. Koperska** [1,*] , **Jacek Bagniuk** [1], **Małgorzata M. Zaitz-Olsza** [1], **Katarzyna Gassowska** [1], **Dominika Pawcenis** [1], **Maciej Sitarz** [2] , **Ewa Bulska** [3] **and Joanna Profic-Paczkowska** [1]

[1] Faculty of Chemistry, Jagiellonian University, Gronostajowa 2, 30-387 Cracow, Poland
[2] Faculty of Materials Science and Ceramics, AGH University of Science and Technology, A. Mickiewicza 30, 30-059 Cracow, Poland
[3] Biological and Chemical Research Centre, Faculty of Chemistry, University of Warsaw, Żwirki i Wigury 101, 02-089 Warsaw, Poland
[*] Correspondence: mkoperska@gmail.com

**Featured Application: For heritage science purposes, especially to preserve silk artifacts.**

**Abstract:** This study investigates the degradation mechanism of silk fibroin through Fourier-transformed infrared spectroscopy (FTIR) analysis. The secondary structure of silk fibroin-based materials is monitored using FTIR, and various estimators are calculated to assess the impact of degradation conditions and aging time. The oxidation estimator shows consistent growth, indicating peptide bond oxidation from the early stages of artificial aging, regardless of the conditions. The environment influences the hydrolysis estimator, with water introduction leading to significant changes. The crystallinity estimator reflects the overall degradation level, affected by oxidation and hydrolysis. XRD and FTIR analysis of historical silk banners up to 500 years old demonstrate a decrease in crystallinity and an increase in hydrolysis and oxidation. The presence of water accelerates the oxidation process, while crystallinity changes are primarily driven by oxidation. Fibroin degradation affects both antiparallel and parallel regions, with water playing a crucial role in accelerating hydrolysis and causing structural shifts. This study enhances our understanding of silk fibroin degradation and provides valuable insights for preserving historical silk artifacts.

**Keywords:** silk; fibroin; pol-ATR; in situ ATR FTIR; crystallinity estimators; artificial aging

## 1. Introduction

The primary motivation behind our study was the urgent need for conservation intervention on valuable 16th-century silk banners stored at the Wawel Royal Castle in Cracow, Poland [1]. We saw this as an opportunity to evaluate the reliability of FTIR-derived estimators and determine if similar trends in estimator values are observed under different thermal aging conditions. Thus, we aimed to ascertain whether artificial and natural aging can be compared using the same FTIR-derived estimators or if more invasive analyses are required to assess the degradation state of the museum objects.

The side reason for our work is to investigate whether the accelerated aging experiments can aid in understanding historical samples' natural degradation and behavior. These experiments aim to simulate the effects of natural aging by subjecting the samples to accelerated conditions, including temperature, Volatile Organic Compounds (VOC), and humidity [2]. These controlled stimuli are expected to induce changes that mirror natural degradation reactions.

Heritage science, a multidisciplinary field merging science and art, plays a pivotal role here. It enables informed decisions regarding the preservation of artifacts through comprehensive physicochemical analyses of various materials (including *Bombyx mori* silk). These analyses help determine degradation mechanisms and provide historical

information about heritage objects' composition, handling, and manufacturing processes [2]. Understanding the Fourier-transform infrared (FTIR) spectra of deteriorated silk fibroin samples is crucial. FTIR analysis can be conducted without damaging the sample and, in some cases, without extensive sampling procedures (if the object can be placed in the ATR spectrometer attachment) [2,3]. Analyzing the FTIR spectra of silk samples has been evaluated for a few daces now due to the complexity of the structure of the samples [3].

*Bombyx mori* silk is a semi-crystalline biopolymer characterized by highly organized nanocrystals enveloped within an amorphous matrix [4]. Fibroin microfibrils, with widths of up to $6.5 \times 105$ nm, are composed of helically packed nanofibrils (ranging from 90 to 170 nm in diameter [2] (see Figure 1)) that consist of heavy and light polypeptides weighing approximately 350 kDa and 25 kDa, respectively [3,4]. The tertiary structure of the protein is stabilized by various interactions, including van der Waals bonds between peptide segments, hydrogen bonds located around 0.5 nm from the -C=O and N-H groups of adjacent peptides, and disulfide crosslinks [5].

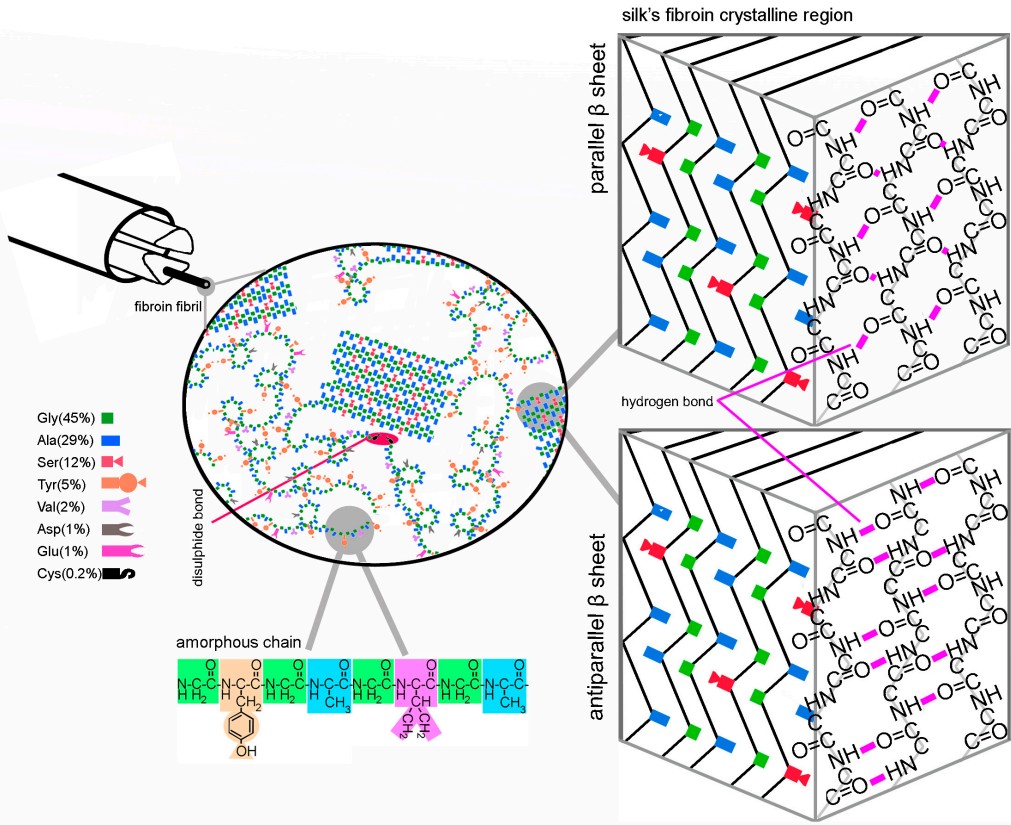

**Figure 1.** Scheme of silk's fibroin primary and secondary structure.

Although various amino acids compose silk fibroin, its crystalline regions consist of patterns composed solely of the simplest amino acids, such as (-Gly-Ala-Gly-Ala-Gly-Ser-Gly-Ala-Ala-Gly-[Ser-Gly-(Ala-Gly)$_2$]$^8$-Tyr) [5]. Small functional groups, such as methyl or hydroxyl, allow for the close alignment of amino acid chains, facilitating the formation of β-sheet secondary structures in silk's fibroin crystalline regions (see Figure 1). Van de Waals interactions strongly stabilize the packed structure of fibroin β-sheets. The distances between pleats are 3.7 Å/5.7 Å (the inter-sheet alternating distances), and distances within pleats are 4.4 Å (linked with the intra-sheet interchain distance) [5]. Crystalline regions are surrounded by disordered chains enriched with heavier amino acids, known as amorphous regions. The ordered regions contribute to silk's characteristic strength, while the non-ordered regions provide resilience [2].

In heritage science, silk samples undergo natural aging, making it equally important to comprehend the model sample's structure and degradation pathway [2,3]. Artificial

aging is primarily carried out through temperature and light interaction to gain insights into the decomposition mechanism of model samples [3]. This paper specifically focuses on the artificial aging induced by thermal conditions in silk samples.

The initial attempts to describe the long-term degradation of fibroin due to high temperatures can be traced back to 1989. During a presentation for the American Chemical Society, Becker demonstrated that a temperature of 150 °C was the most suitable for fibroin degradation [6]. It is important to note that his experiments were preliminary and focused on various textile and paper materials. Since then, infrared spectroscopic research on the long-term high-temperature degradation of fibroin has primarily focused on investigating the vibrations of C=O and N-H in the spectra's Amide I and II regions [7–9]. It has been shown that this can mainly assess the oxidation of fibroin [1]. In our previous work, we have proposed to look in the $\delta_{C-H}$ region of FTIR spectra and have put forward the hydrolysis estimator as integral of 1318 cm$^{-1}$ band, which corresponds to deformation vibrations of groups in a free amino acid, or symmetric bending found in free dicarboxylic amino acids (COO-NH-...-COOH) to band integral of CH$_3$ bending vibration band located at 1442 cm$^{-1}$ [1,10–12].

However, the most extensively studied estimator in the literature using infrared techniques is the crystallinity estimator (often confirmed by XRD analysis, as XRD is one of the primary methods used to evaluate the degree of crystallinity in silk samples [13]. In FTIR, the ratio of two Amide I band intensities at 1615 and 1655 cm$^{-1}$ (reflecting the relative proportion of the polymer in an organized β-sheet, α-helix, or random coil arrangement) and the ratio of Amide III bands at 1264 and 1230 cm$^{-1}$ can be utilized to calculate the crystallinity index based on similar principles [5,8,14–19]. Alternatively, the growth of the Amide I sub-band centered at 1699 cm$^{-1}$ can be monitored [20,21]. Literature reports indicate that a decrease in crystallinity is often associated with a decline in handling properties, making crystallinity measurement valuable for conservators and museum professionals worldwide [8,22]. Various attempts have been made to measure the degree of crystallinity in historical silk samples using FTIR and XRD techniques [6,19,23–27]. Zhang et al., analyzed XRD patterns and FTIR spectra, revealing fluctuations in crystallinity in ancient samples [9,28]. More detailed findings from Greiff et al., indicate the continued presence of crystalline domains in aged samples, although they exhibit almost complete disorientation in historical samples [29]. In wide-angle X-ray scattering (WAXS) experiments conducted by Martel et al., and ATR-FTIR studies by Arai et al., the loss of the amorphous phase has been identified as a cause of reduced mechanical strength of silk fibers [23,30]. However, Hermes and Liu reported difficulties obtaining XRD reflections from ancient samples [17,19]. It is essential to acknowledge that the estimation of crystallinity from FTIR spectra can sometimes be distorted by vibrations, such as those from oxidation products that occur during the degradation process, leading to changes in the shape of the Amide bonds and complicating calculations of crystallinity changes.

Our previous work was dedicated to studying the degradation process and the change in estimators during thermal degradation. We validated the estimators derived from FTIR measurements using XRD, size exclusion chromatography (SEC), and ultraviolet-visible spectroscopy (UV-Vis). We specifically investigated the oxidation estimator (validated with UV-Vis spectra [1] and color change [31]), the hydrolysis estimator (confirmed with SEC analysis [1]), and three crystallinity estimators. However, only one of the crystallinity estimators showed a similar change trend to the XRD calculations [9]. Consequently, we identified three promising degradation markers, which we used to assess the condition of 10 historical samples dating from the 16th to 19th century [32]. The correspondence of oxidation and hydrolysis estimators and the object's manufacturing origin or production period was discussed. However, the crystallinity estimator showed inconsistent trends over time, raising questions about the nature of changes during artificial and natural aging processes. This discrepancy prompted us to investigate the changes in crystallinity degree during aging. In this publication, we have focused on thermally-induced degradation using XRD and ATR-FTIR techniques to analyze fibroin's crystallinity degree degradation

process. However, we aimed to observe degradation in situ to ensure data reliability without altering test spots over time.

## 2. Materials and Methods

### 2.1. Model Silk Sample

Bleached and degummed *Bombyx mori* silk (35 g/m$^2$), manufactured for conservation use [2], was purchased from a Chinese retailer (Sailong, Warsaw, Poland) studied previously [1]. It serves here as a model silk sample (MS).

### 2.2. Historical Silk Samples

Samples from three historic silk banners were collected thanks to the cooperation with the Museum of Wawel Castle in Cracow, Poland (samples described elsewhere) [31]. The probed museum objects included an Inscription Banner from the 17th century (IB, see Figure 2A); a Court Banner from the 16th century (CB, see Figure 2B); and Stanisław Barzi's Funeral Banner also from the 16th century (FB, see Figure 2C). They are depicted in Figure 2 and listed in Table 1.

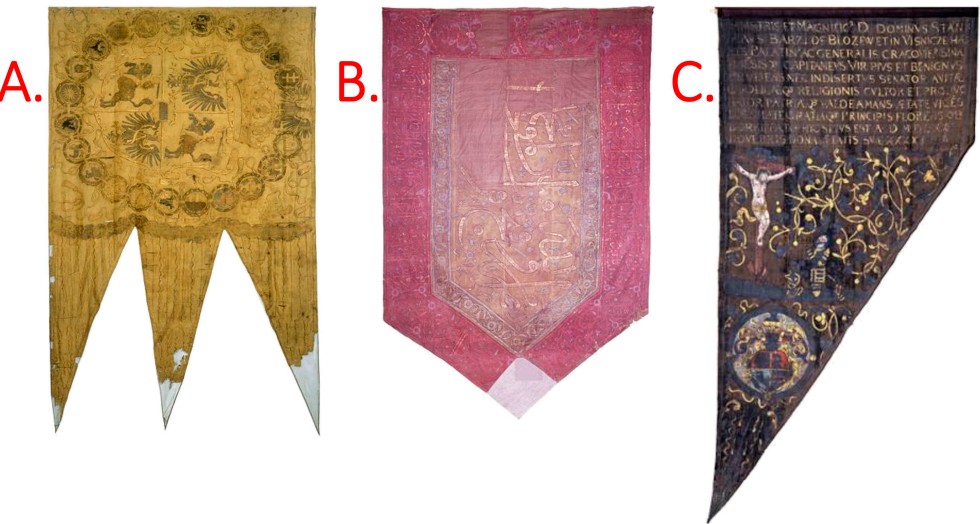

**Figure 2.** Historic banners from the Museum of Wawel Castle collection: (**A**) Inscription Banner 17th century, Turkey (IB); (**B**) Court Banner 16th century, Poland (CB); (**C**) Stanisław Barzi's Funeral Banner 16th century, Poland (FB).

**Table 1.** Analyzed silk samples: short name (abbreviation used in this work), full name, aging factor, maximum time, and whether it is a museum sample or model one.

| Sample—Short Name | Sample—Full Name | Ageing Factor/ | Ageing Time | Museum (Century)/Model Sample |
|---|---|---|---|---|
| FB | Stanisław Barzi's Funeral Banner | natural | 332 years | museum (16th) |
| CB | Court Banner | natural | 462 years | museum (16th) |
| IB | Inscription Banner | natural | 485 years | museum (17th) |
| MS | model silk sample | none | t = 0 | model |
| esO$_2$ | ex situ aged in air/open | thermo-aged/O$_2$ | up to 14 days | model |
| esO$_2$/VOC/H$_2$O | ex situ aged in air/closed | thermo O$_2$/VOC/H$_2$O | up to 14 days | model |
| isO$_2$ | in situ aged in air/open | thermo-aged/O$_2$ | up to t = 120 min | model |

### 2.3. Artificial Ageing

This paper employed two methods of artificial thermal aging: in situ and ex situ. In the in situ approach, spectroscopic measurements were performed on silk samples during the aging process at the sampling spot (is$O_2$). On the other hand, the ex situ method involved aging the samples under specific conditions and subsequently analyzing them separately (es$O_2$, es$O_2$/VOC/$H_2O$). One significant distinction between these methods is the time scale, with the in situ experiments lasting approximately 12 h and the ex situ experiments lasting up to 14 days. Due to the variation in sampling spots, the characterization of ex situ samples will result in different errors due to statistical variations. Furthermore, the different conditions employed in each method may activate distinct degradation pathways, with hydrolysis, for instance, not occurring during the in situ experiments.

For ex situ aging tests, silk samples (c.a. 0.07 g) were hung in the ventilated aging (climatic) chamber and closed in vials (c.a. 140 mL) in different content of the gaseous atmosphere and kept at the temperature of 150 °C for up to 14 days [1]. In this way, the influence of different agents on the degradation of silk was traced down:

A. In a climatic chamber maintained at dry conditions (es$O_2$), oxygen's effects were used to follow fibroin's oxidation.
B. In closed vials, oxygen, water vapor, and gaseous degradation products (called volatile organic compounds (VOC)) were called for to simulate the most severe degradation, including oxidation, hydrolysis, and autocatalytic effects (es$O_2$/VOC/$H_2O$).

In situ aging experiments were performed at the same temperature but lasted a maximum of 12 h with the aid of:

- An in-build heated board, which triggered the thermo-degradation from the silk's surface, was placed in a geometry allowing that surface to be measured during aging with ATR-FTIR.
- Gas flow system that allowed to flush of the samples with a given gas mixture containing oxygen to follow oxidation (is$O_2$) to focus more on crystallinity changes.

### 2.4. Spectroscopic Ex Situ Analysis

X-ray diffraction (XRD) analysis was performed using Bruker D2 Phaser X-ray powder diffractometer (30 kV, 10 mA) equipped with an LYNXEYE detector using Cu K$\alpha$ radiation (0.15418 nm at room temperature). The patterns were collected for the same set of samples as for the FTIR measurements, with a step size of 0.02 and scan rate of 1 s in the 2θ range of 5–40; repeated once due to large sampling spot size (about 15 mm). The diffractograms were resolved by fitting the six Gaussian curves and calculating the crystallinity estimator as a quotient of integrals of peaks attributed to the β-sheet region to the rest of the peaks [23].

Infrared spectra were collected employing the Fourier Transformed Infrared Thermo Nicolet 8700 spectrometer with an MCT-B detector equipped with a Pike GladiATR attenuated total reflection accessory with a diamond crystal and heated table. The spectra were collected in the 4000–650 cm$^{-1}$ range with an optical resolution of 4 cm$^{-1}$, averaging 200 scans. Predefined and constant pressure between the sample and the ATR window was ensured. For each sample, three spectra were acquired. Each spectrum was corrected using automated ATR correction [9].

Before ATR-FTIR measurements, samples aged 1 to 14 days were dried at 110 °C for 10 min to remove the water bonded to the silk structure that would shield the vibrations of Amide I and other carbonyl groups evolving upon oxidation and hydrolysis. For this purpose, a heating ATR stage was used.

### 2.5. In Situ Spectroscopic Analysis

In situ, FTIR spectra were collected utilizing the same spectrometer and detector as in ex situ studies, equipped with Pike GladiATR Attenuated Total Reflection (ATR) device with a diamond crystal and an in-build, heated, stainless-steel plate with temperature regulation up to 250 °C. Here predefined and constant pressure between the sample and

the ATR window was kept during 12 h of aging and measuring. FTIR spectra were collected, averaging 200 scans about every 80 s of 720 min, repeated thrice, and averaged.

### 2.6. pol-ATR-FTIR

Polarisation spectroscopy uses linearly polarised light instead of an unpolarised light beam. GladiATR attenuated total reflection (ATR) device was equipped with a polarising filter (pol) with an adjustable angle θ. Spectral measurements were carried out at different angles θ between the plane of polarization of the light and the axis of the sample [33]. For materials with a partially ordered structure, quenching and amplifying some absorption bands may be observed during the change in the θ angle of polarisation of light. This phenomenon is observed because the bond's radiation absorption probability is maximal at a certain angle (between the beam polarisation plane and the axis of the bond) [7]. Pol-ATR-FTIR spectra were gathered with the angle step of 45°, in the range of 0–360°, and 15° step, in the range of 0–90°.

### 2.7. Estimator Definitions

All spectra were transferred to the Origin program, where they underwent treatment meaning the data for all the defined below estimators were gathered there. Thus, it allowed calculating estimators to concentrate on the secondary structure of the protein—the crystallinity estimator. All of them can be found in the literature and are defined as follows:

- From ATR-FTIR spectra, the $E_C$—intensity ratios within Amide I C=O stretching vibration of parallel β-sheet to antiparallel β-sheet A1620/A1699 [21]). Absorption values were calculated with a straight-line baseline from 1818 to 866 cm$^{-1}$.

- From pol-ATR-FTIR spectra, the $E_{CPOL}$—intensity ratios for this were calculated from polarised spectra recorded at 0° and 90° for each absorption band [8]. Absorption values were calculated from Equation (1) with a straight-line baseline from 1818 to 866 cm$^{-1}$.

$$A_{POL} = (A\| + 2A\perp)/3 \tag{1}$$

- From the XRD pattern, the $E_{\beta XRD}$—is the sum of fitted peak areas of β-sheet domains divided by the sum of all fitted peaks [34]. Peak analysis was performed on spectra with a polygonal-line baseline from 10 to 35°.

It is essential to highlight that several methods of calculating crystallinity index and beta content can be found in the literature. Authors use three to six Gaussians to fit the XRD patterns [23,35]. Before calculating $E_{\beta XRD}$, diffractograms (average ones calculated from three separate measurements) were resolved by fitting six Gaussian curves. No baseline subtraction was needed. These Gaussian curves were then attributed to silk I (two peaks centered at 19.22° (4.6 Å) and 20.42° (4.3 Å) for the MS sample), from silk II (three peaks at 24.82° (3.6 Å), 27.98° (3.2 Å) and 30.53° (2.9 Å)) and amorphous domains (21.89° with average distances of 4.1 Å). All six peaks were found in MS and artificially aged and historical samples. Fitting was performed with an r2 value of 0.999 (+/−0.001).

Additionally, in the spectra collected by ATR-FTIR, the primary functional groups' estimators were monitored to give more insight into the provoked degradation pathways. We decided to use estimators proposed/defined and evaluated in our previous work [8,33], which are:

- Oxidation estimator: $E_{AmideI/II}$—intensity ratios of Amide I C=O stretching vibration to Amide II N-H in-plane bending and C-N stretching vibrations $A_{1620}/A_{1514}$ (baseline as above).

- Hydrolysis estimator: $E_{COOH}$—band 1318 cm$^{-1}$ integral to band integral of CH3 bending vibration band located at 1442 cm$^{-1}$ $P_{1318}/P_{1442}$ (baseline as above).

Pol-ATR-FTIR allowed calculating dichroic ratio *R* (defined in Equation (2)) and order parameter *S* (defined in Equation (3))

$$R = A\|/A\perp \tag{2}$$

$$S = \frac{R - 1}{R + 2} \tag{3}$$

The *R* parameter allows us to estimate the degree of dichroism; *R* = 1 means no dichroism, and *R* = 0 or to infinity, respectively, represents the maximum dichroism for completely parallel or perpendicular orientation of IR-activated transition moments to the molecule's axis. The *S* parameter investigates the degree of order in the sample's secondary structure sub-domains; it is set to 0 for the disordered structure. Depending on the same moments of transition geometry to the molecule's axis, it equals 1 when they are parallel and −0.5 when perpendicular [8,36,37].

## 3. Results

In our previous studies [9,33], we investigated the secondary structure of silk. Our goal was to derive degradation estimators from the FTIR spectra. We proposed estimators calculated for oxidation ($E_{AmideI/II}$), hydrolysis ($E_{COOH}$), and crystallinity ($E_C$) based on the spectra. We validated their chemical interpretation through UV-Vis, SEC, and XRD analyses, respectively. We have compared in situ and ex situ artificial aging with naturally aged historical samples using these IR-derived estimators to analyze the FTIR data further. Additionally, we verified the crystallinity estimator by comparing it with XRD analysis.

### 3.1. Short-Time Degradation Viewed by In Situ Analysis on Model Samples

In the in situ thermo-aging experiments, we focused on the initial stages of fibroin degradation. We recorded FTIR spectra in the spot where degradation was occurring, induced by high temperature, under dry airflow conditions (isO$_2$) (refer to Figure 3B). The $E_{AmideI/II}$, $E_{COOH}$, and $E_C$ estimators, previously established during long-term ex situ FTIR experiments, proved effective in assessing the state of fibroin (thermal) degradation. This allowed us to observe and describe the short-term in situ degradation process. However, the data obtained during the first hour of the in situ experiments are irrelevant, likely due to the sample reaching thermal equilibrium.

The $E_{COOH}$ estimator, representing hydrolysis, exhibited observable changes indicating the ongoing hydrolysis process within silk's fibroin. These changes were identified by developing vibration bands related to dicarboxylic amino acids. Specifically, the band centered at 1318 cm$^{-1}$ corresponds to -C-H symmetric bending found in free dicarboxylic amino acids (COOH-NH-...-COOH) [10,12]. In ex situ artificial aging under open dry conditions (isO$_2$), this hydrolysis process was minimal (with a max value around 0.05), with a relatively sizeable statistical error. However, for the short-term in situ aging, the change in the estimator was approximately one order of magnitude smaller (depicted by the purple line in Figure 3B, based on an average of 3 measurements). The hydrolysis process was not triggered in the short-term in situ aging conditions, possibly due to a lack of water access in the pressed region of the ATR-FTIR module where the silk model sample was aged.

The $E_{AmideI/II}$ estimator (depicted by the pink line in Figure 3B) exhibited a steady increase over time, observed in both ex situ and in situ aging. Like the $E_{COOH}$ estimator for in situ aging, the changes during short-term artificial aging were relatively small. However, this growth was attributed to the overall increase in C=O groups resulting from temperature-induced oxidation. These groups could manifest as ketones, aldehydes, or carboxylic groups in *α*-keto acids and dicarboxylic groups, as described in previous studies [1].

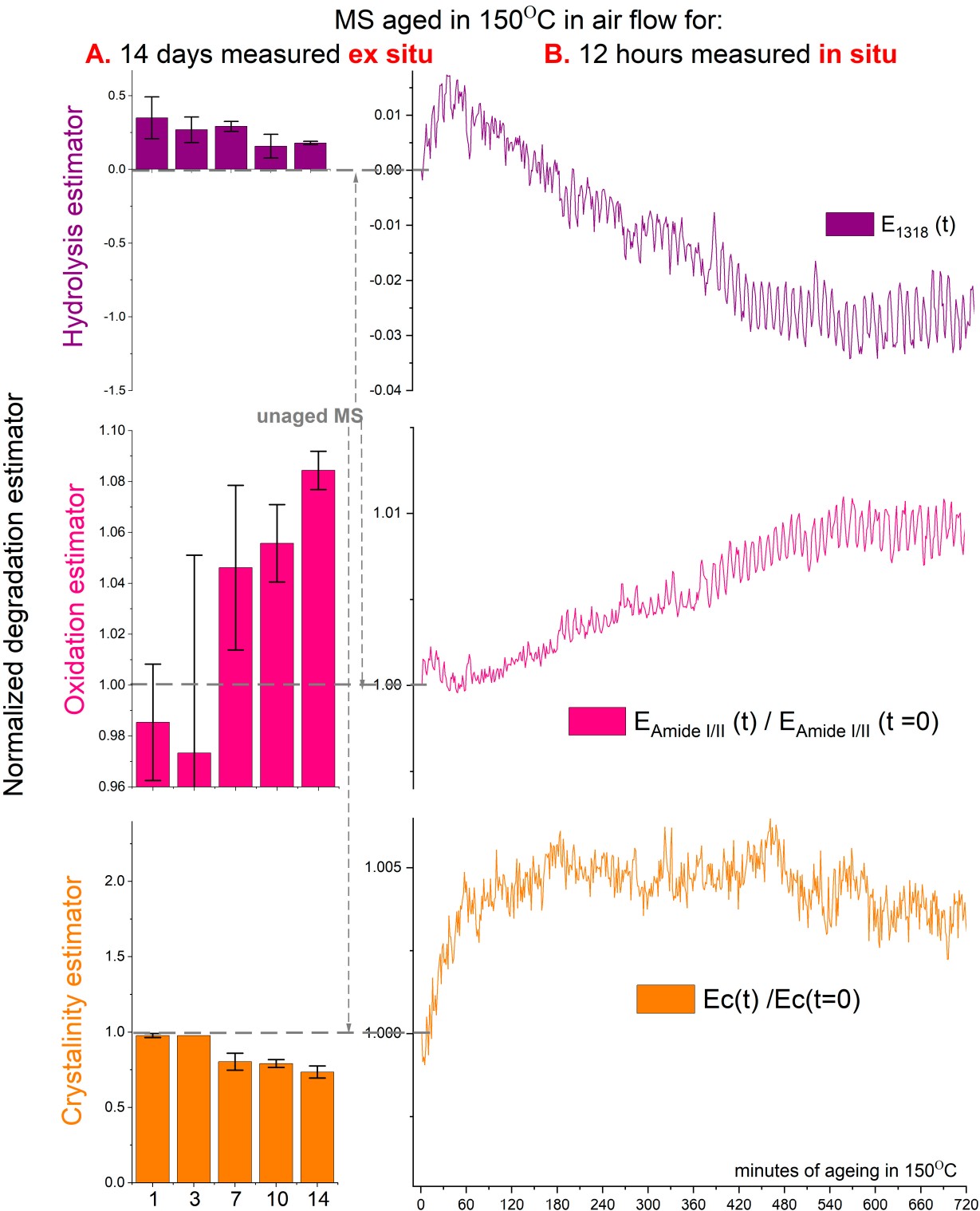

**Figure 3.** Hydrolysis, oxidation, and crystallinity estimators calculated from ATR-FTIR for (**A**) ex situ thermo-aged MS in ventilated reactor esO$_2$ aged for 1, 3, 7, 10, and 14 days, and (**B**) in situ thermo-aged MS in air XRD/isO$_2$ for 12 h (example of one of the three averaged measurement trends, an average of 3 measurements in red).

The change In the E$_C$ estimator recorded in situ (depicted by the orange line in Figure 3B) revealed a continuous increase, reaching a plateau in the third hour of artificial aging, followed by a slower and more gradual decline. The slight initial growth of the

estimator (compared to ex situ changes in Figure 2A) within the first hour of aging may be attributed to accumulated heat energy on the artificially aged spot, potentially causing short-term structural relocations within the semi-crystalline structure of silk. However, this should be considered insignificant. The changes in the $E_C$ estimator resulting from short-term in situ aging were an order of magnitude smaller than those observed in ex situ aging conditions. Consequently, FTIR analysis detected no significant changes in the degree of crystallinity during this short-term aging simulation.

### 3.2. Long-Time Artificial Ageing Vied by Ex Situ Analysis on Model Samples

Our previous work [9] provided detailed studies on long-term artificial aging conducted ex situ under various conditions, including closed and open reactors with both humid and dry environments. By examining the changes in estimators, we gained insights into the mechanism of artificial degradation over an extended period induced by high temperatures and analyzed ex situ.

The $E_{COOH}$ hydrolysis estimator demonstrated some inconsistent growth during long-term ex situ degradation (depicted in pink in Figure 3A), with maximum values around 0.18 for samples aged in dry conditions $esO_2$ comparable to those calculated for naturally aged samples (from around 0.6 to 2.1). This trend is not evident in the ex situ estimator, which decreases after 14 days compared to the increase observed within the first day.

The $E_{AmideI/II}$ oxidation estimator (depicted in purple in Figure 3A) steadily increases, with changes reaching a significant magnitude of approximately 1.2. Although smaller, this increase is noteworthy compared to the corresponding values for historically aged samples (ranging from 1.3 to 1.8). The oxidation process progresses consistently throughout the long-term ex situ aging.

Simultaneously, the crystallinity estimator $E_C$ (depicted in orange in Figure 3A) consistently decreases during ex situ aging. In these artificial aging conditions, the loss of crystallinity leads to the formation of less organized silk structures, which may eventually network with each other, as reported in gel conditions [38,39].

### 3.3. Super-Long-Time Natural Ageing: Historical Banners Analysis by FTIR

Our previous results [31] (Figure 4) focused on the estimator values obtained from historical textiles, explicitly selecting the three most valuable and oldest examples for comparison with artificially aged model silk samples. The estimators discussed below are $E_{COOH}$, $E_{AmideI/II}$, and $E_C$.

The hydrolysis estimator ($E_{COOH}$) demonstrates a growing trend in older historical samples. However, it is essential to note that the unusually high value observed for the Inscription Banner (IB) could be attributed to the origin of the silk sample. The IB was produced in the Ottoman Empire, while the rest of the banners were manufactured in Europe. Nonetheless, the high $E_{COOH}$ values indicate that older silk samples exhibit more peptide bond cuts compared to the ongoing networking effect.

The oxidation estimator ($E_{AmideI/II}$) increases over time and consistently exhibits higher values in older samples. This indicates that the presence of oxygen contributes to the oxidation of silk. At the same time, the hydrolysis of peptides and changes in their ordering does not significantly impact the oxidation process.

It is crucial to consider that the high oxidation levels of peptide bonds in historical samples can influence the Amide I region of IR spectra, leading to distortions in the values of the crystallinity estimator $E_C$. We have observed drops in $E_C$ values for very old historical samples, although this observation is consistent only for certain analyzed textile artifacts. In some naturally aged silk, higher crystallinity can be observed, aligning with findings reported by other researchers [29,40]. The faster decomposition of amorphous regions in the form of VOC may also explain the relative increase in $E_C$ during degradation. Moreover, potential crosslinking may occur, forming hydrogen bonds between carbonyls and -NH/-OH groups, allowing for slight recrystallization. These factors can contribute to a relative rise in calculated crystallinity from IR spectra or at least hinder its accurate

determination. However, further analysis using pol-ATR-FTIR was conducted primarily to explore the possibility of recrystallization to explain the above.

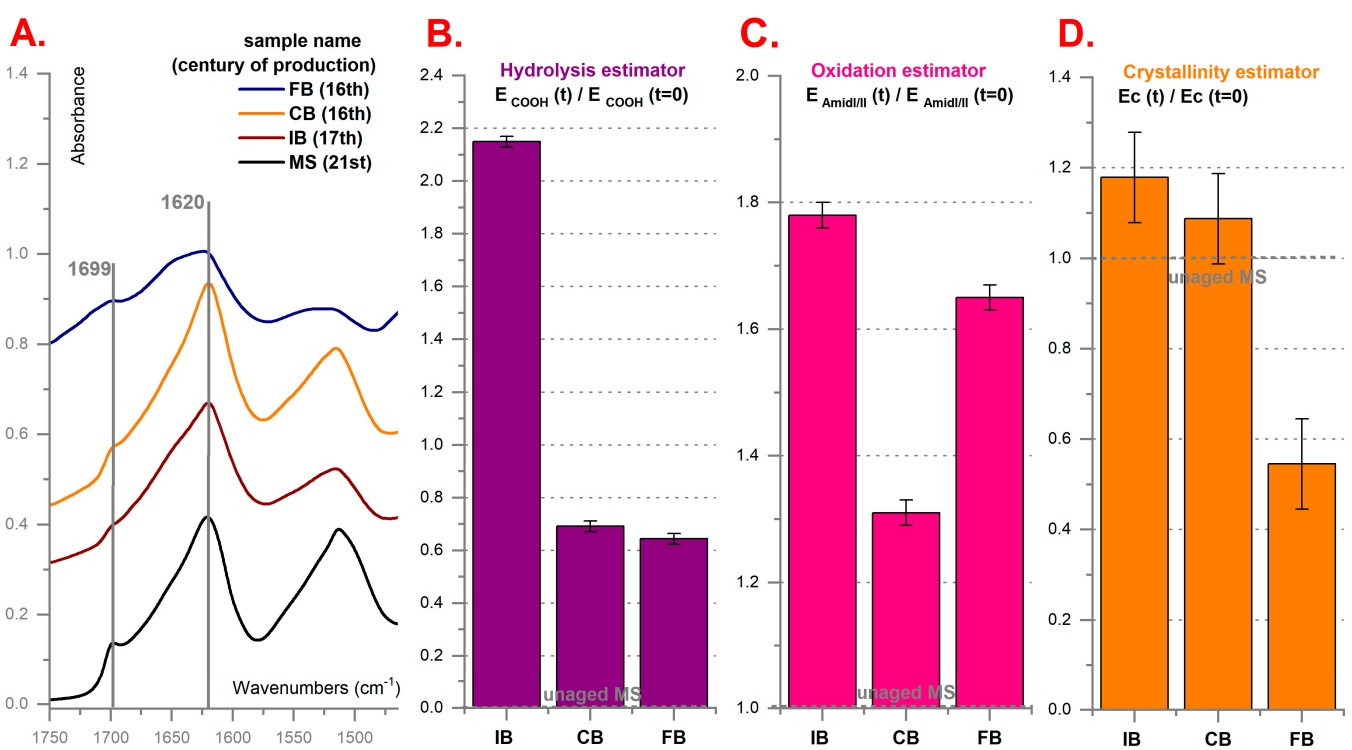

**Figure 4.** (**A**) ATR-FTIR spectra and normalized fibroin (**B**) Hydrolysis, (**C**) Oxidation and (**D**) Crystallinity estimators calculated from ATR FT-IR for model silk sample (MS) and historical: Inscription Banner (IB); Court Banner (CB); Stanisław Barzi's Funeral Banner (FB).

*3.4. Detailed Analysis of FTIR Crystallinity Estimator at Different Polarisation Angles*

Pol-ATR-FTIR experiments were performed to obtain a better understanding of FTIR crystallinity estimators. Pol-ATR-FTIR spectra of a few fibroin fibers obtained with the linearly polarised light at different angles from 0° (A∥, parallel spectra) to 90° (perpendicular spectrum, A⊥) are depicted in Figure 5. Amide I band was particularly interesting in these spectra, composed of parallel and perpendicular dichroism due to coupled modes in beta-sheets. In this vibration region, coupling between four peptide bonds is expected to result in four normal modes, out of which two are active and strong enough to be observed on IR spectra: $v\|(0, \pi)$ and $v\perp(\pi, 0)$ [31]. Centered at 1699 cm$^{-1}$ $v\|(0, \pi)$ band reflects the situation where two adjacent amide groups in the same chain vibrations are in-phase but are out-of-phase with adjacent chains amide groups (in antiparallel pleats), whereas centered at 1620 cm$^{-1}$ $v\perp(\pi, 0)$ band corresponds to the reversed situation where within the same chain amide groups vibe out-of-phase but in-phase with neighboring chains groups (in parallel pleats) [33]. Superimposed over these bands is one cantered at 1656 cm$^{-1}$ that describes oscillations of amide groups in α-helix or random coil parts of spectra. Thus pol-ATR-FTIR spectra, recorded using a manually operated polarizer, show that changing polarisation states of light results in different absorption coefficients for MS sample absorption bands at distinct wavelengths (see Figure 4). Thus, the Amide I band is most intensive at 0° and Amide II at 90° with intermediate states at remaining angles.

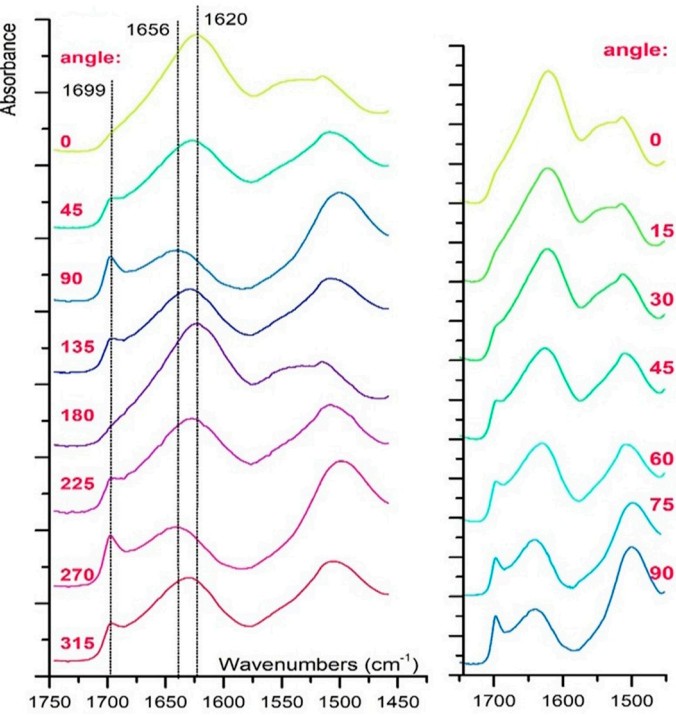

**Figure 5.** Pol-ATR-FTIR spectra of silk MS recorded with polarizer angles of 315°, 0°, 45°, 90°, and 135°.

Bands from the Amide I region were carefully analyzed, and the dichroic ratio $R$ and order parameter $S$ was calculated for sub-bands centered at 1699, 1956, and 1620 cm$^{-1}$. A significant degree of dichroism is observed for the bands referenced to β-sheet ordered regions (1699 and 1620 cm$^{-1}$). $R$ values around 3.322 and 0.352 (for 1699 and 1620 cm$^{-1}$, respectively) suggest a high degree of orientation to the fiber axis of IR activated transition dipole moment of the given vibration band (see Table 2). Crystalline regions vibrations show some ordered structure with values of orientation parameter $S$ at 0.427 and −0.277, confirming in-phase and out-of-phase vibration corresponding to 1699 cm$^{-1}$ as parallel and 1620 cm$^{-1}$ as perpendicular to the crystalline domain axis [37]. Little dichroism is attributed to 1656 cm$^{-1}$, with the $R$-value close to 1 and $S$-value close to 0, as this is a band associated mainly with stretching vibration of C=O bond located in random coil and α-helix domains [33].

**Table 2.** Values of parameters $R$ and $S$ calculated for three different wavelengths from the collection of pol-ATR-FTIR for the initial model silk sample (MS).

|  | $R$ | $S$ |
|---|---|---|
| 1620 cm$^{-1}$ | $0.352 \pm 0.010$ | $-0.277 \pm 9.67 \times 10^{-5}$ |
| 1656 cm$^{-1}$ | $0.718 \pm 0.016$ | $-0.104 \pm 2.43 \times 10^{-4}$ |
| 1699 cm$^{-1}$ | $3.322 \pm 0.122$ | $0.427 \pm 1.49 \times 10^{-2}$ |

According to the dichroic ratio $R$ and order parameter $S$-values presented in Table 3 and Figure 6, the highly ordered orientation of silk vibrations in the crystalline domain changes artificial and natural aging conditions.

**Table 3.** Dichroic ratio *R* and order parameter *S* calculated from pol-ATR-FTIR for two bands in the superimposed Amide I region of the spectra (bands centered at 1620 and 1699 cm$^{-1}$).

| Sample's Name | MS | | isO$_2$ | | esO$_2$ | | esO$_2$/VOC/H$_2$O | |
|---|---|---|---|---|---|---|---|---|
| | **1620** | **1699** | **1620** | **1699** | **1620** | **1699** | **1620** | **1699** |
| *R* | 0.368 | 3.005 | 0.438 | 1.572 | 0.486 | 1.776 | 0.751 | 0.675 |
| (+/−) | 0.027 | 0.095 | 0.013 | 0.012 | 0.023 | 0.109 | 0.014 | 0.001 |
| *S* | −0.270 | 0.393 | −0.232 | 0.160 | −0.205 | 0.212 | −0.091 | −0.122 |
| **Sample's name** | **MS** | | **IB** | | **CB** | | **FB** | |
| | **1620** | **1699** | **1620** | **1699** | **1620** | **1699** | **1620** | **1699** |
| *R* | 0.368 | 3.005 | 0.493 | 0.792 | 0.517 | 0.865 | 0.692 | 0.898 |
| (+/−) | 0.027 | 0.095 | 0.007 | 0.033 | 0.005 | 0.020 | 0.014 | 0.019 |
| *S* | −0.270 | 0.393 | −0.203 | −0.074 | −0.192 | −0.047 | −0.115 | −0.035 |

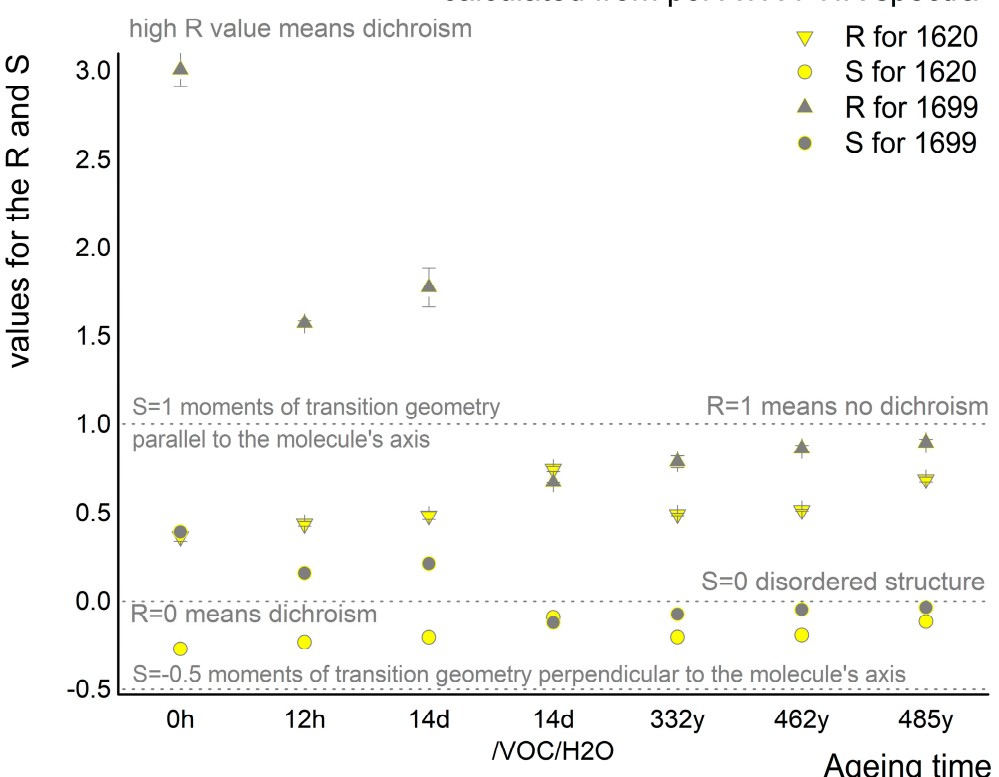

**Figure 6.** Dichroic ratio *R* and order parameter *S* calculated from pol-ATR-FTIR for two bands in the superimposed Amide I region of the spectra (bands centered at 1620 and 1699 cm$^{-1}$).

The *R*-value significantly decreases for antiparallel crystalline domains but increases for parallel vibration orientations. Both regions are undergoing decomposition. Additionally, the antiparallel arrangements seem to undergo structural changes, as only degradation products corresponding to parallel arrangements are present in historical samples and samples aged under harsh conditions (esO$_2$/VOC/H$_2$O).

The changes in *R*-values coincide with the order parameter *S*-values approaching zero during aging, indicating a decrease in ordering in both parallel and antiparallel crystalline pleats. Similarly to *R*-values, the *S* parameter drops below zero for samples aged under the harshest conditions (esO$_2$/VOC/H$_2$O) and for naturally aged historical samples. This suggests a transition to more parallel moments of transition geometry with vibrations becoming more perpendicular to the molecule's axis but different from parallel

pleats. In the antiparallel domains, two adjacent amide groups in the same chain are in-phase, while they are out-of-phase in the adjoining chain, similar to the parallel crystalline domain arrangement. This change could be attributed to the transformation of amino acids from L-amino acids to D-amino acids over time, as recommended by Moini et al., for the Smithsonian Museum as a silk age calculator [41]. The flip between enantiomers in the crystalline regions may explain the observed changes in *S* and *R*-values, which are likely to decompose in this manner due to the overall behavior of D-peptides. Lotz B.in his even proposes that introducing D-enantiomers into the crystalline structure can lead to the disappearance of the pleated structure in favor of rippled one instead of similar structures in nylons [42].

As demonstrated above, the polarizer provides a more comprehensive description of fibroin's secondary structure through FTIR spectra. Since it affects the IR absorption coefficient, it also influences the estimators calculated from the spectra. Figure 7 illustrates that the non-normalized crystalline estimator $E_C$ value changes significantly depending on the polarization angle. The highest value of the estimator is recorded at 0°, while the smallest is at 90°. Therefore, for calculating the $E_{CPOL}$ estimator, an approach based on the theoretical calculations proposed by Boulet-Audet et al., was adopted, and absorption values were computed using both spectra [33].

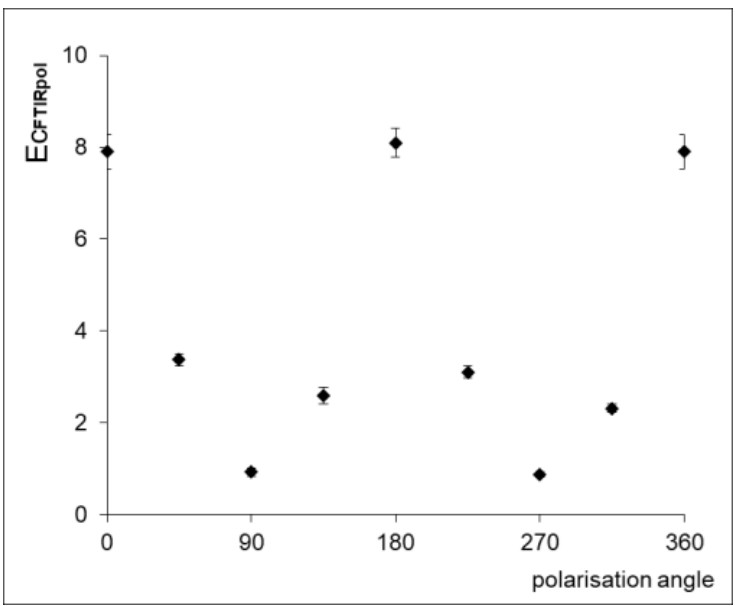

**Figure 7.** Crystalline estimator $Ec_{POL}$ calculated from the pol-ATR-FTIR spectra of silk MS sample recorded with polarizer angles of 0°, 45°, 90°, 135°, 180°, 225°, 270°, 315° and 360°.

The normalized crystalline estimator $E_{CPOL}$, calculated from polarized spectra (in yellow), as compared to the non-polarized estimator Ec (in orange) and depicted in Figure 8. The value of $E_{CPOL}$ is highly affected during the first 12 h of aging, unlike $E_C$. Prolonged artificial aging leads to a continuous decrease in the value of the $E_{CPOL}$ estimator. The drop in both estimators reaches a similar level for the samples aged in vials (es$O_2$/VOC/$H_2O$). $E_{CPOL}$ indicates a decline in relative crystalline content for naturally aged samples, unlike the estimator derived from non-polarized spectra. There is no evidence of recrystallization of the previously ordered fibroin domains. However, the value of $E_{CPOL}$, similar to Ec, does not correlate with the age of historical samples. This does not necessarily imply that crystallinity cannot be considered a degradation estimator for historical objects. Further comparison of the data with those obtained from XRD analysis is necessary.

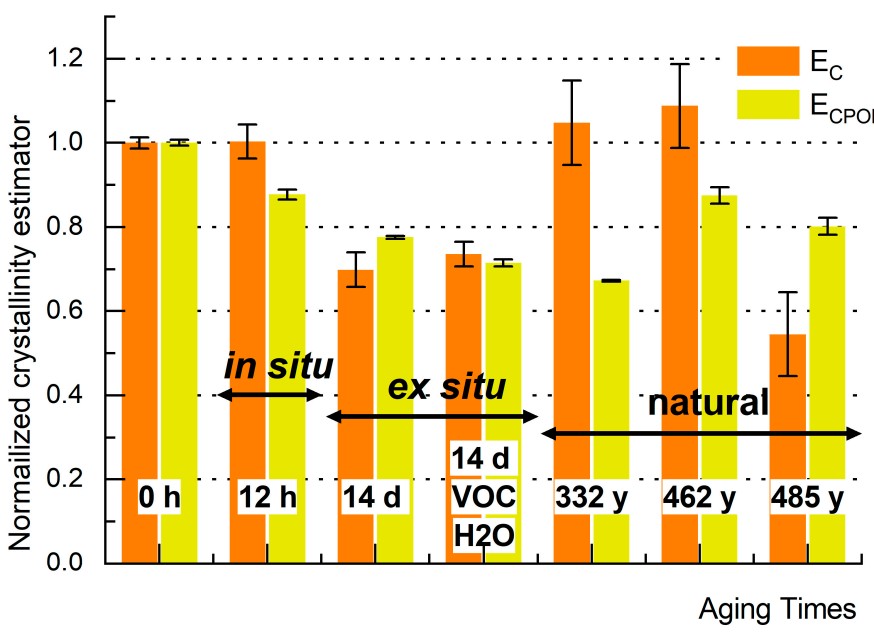

**Figure 8.** Crystalline estimator $E_{CPOL}$ calculated from the pol-ATR-FTIR spectra recorded at polarizer angle 0° (yellow) juxtaposed with Ec, the one obtained from the ATR-FTIR (orange) for model silk sample (MS), in situ thermo-aged MS (isO₂), ex situ thermo-aged MS in the air (esO₂), in the humid closed reactor (esO₂/VOC/H₂O) and historic: Inscription Banner (IB); Court Banner (CB); Stanisław Barzi's Funeral Banner (FB).

## 4. Discussion

To ascertain whether artificial and natural aging can be compared using the same FTIR-derived estimators or if more invasive analyses are required to assess the degradation state of the museum objects results from nondestructive infrared-induced vibration absorption (ATR-FTIR) were compared to the destructive but absolute technique of crystallinity calculation (XRD). Crystallinity estimators obtained from ATR-FTIR ($E_C$ and $E_{CPOL}$) and XRD ($E_{\beta XRD}$) were juxtaposed against the aging time (both artificial and natural). As shown in Figure 9, a similarity is observed in the decreasing trend of both Ec $Ec_{POL}$ and $E_{\beta XRD}$ values for both short-term (in situ) and long-term (ex situ) artificial aging. However, for naturally aged historical samples, the dropping trend of $E_{\beta XRD}$ does not coincide with the estimators derived from FTIR. Unlike artificial aging, the estimators Ec and $E_{CPOL}$ do not exhibit clear trends with aging time or crystallinity content in naturally aged samples. This could be attributed to the overall high oxidation level of the silk sample. The substantial increase in C=O groups due to temperature-induced oxidation may lead to significant changes in the FTIR baseline for the energy regions related to strategic vibrations, causing visible distortions in the derived estimators. Moreover, the number of well-organized regions, as indicated by a dichroic ratio close to 1 and an order parameter close to 0, has significantly diminished in the historical samples. Consequently, the vibrations of these samples might not exhibit strong absorbance in the FTIR spectra. It is reasonable to recommend using FTIR-derived estimators only when the dichroic ratio $R$ is higher than zero, indicating the presence of antiparallel β-sheet pleats in the silk sample. Further investigation will explore the nature of transformations occurring in β-sheets, α-helix, and amorphous regions during both inherent and simulated aging processes.

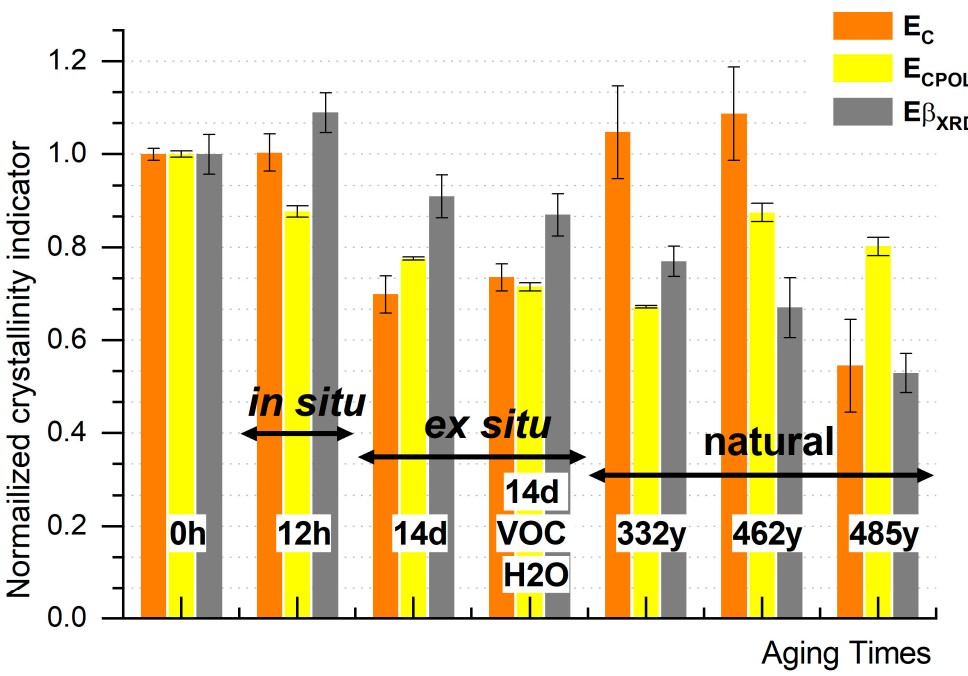

**Figure 9.** Crystallinity ($Ec_{FTIR}$) and crystallinity corrected by polarizer ($Ec_{FTIRpol}$) estimators calculated from ATR-FTIR and β-sheet content estimator ($Eβ_{XRD}$) and crystallinity estimator ($Ec_{XRD}$) derived from XRD for model silk sample (MS), in situ thermo-aged MS in the air ($isO_2$), ex situ thermo-aged MS in the air ($esO_2$), in the humid closed reactor ($esO_2/VOC/H_2O$) and historic: Inscription Banner (IB); Court Banner (CB); Stanisław Barzi's Funeral Banner (FB).

Unlike the crystallinity estimator, the oxidation and hydrolysis FTIR-derived ones can aid in understanding historical samples' natural degradation and behavior by investigating the artificial aging process. As shown in Figure 10, the oxidation estimator exhibits the most consistent trend across different aging methods and durations and thus can be a potential age-correlated estimator. Its value consistently increases from the first hour of in situ artificial aging to the 14 days of all artificial ex situ aging, even in the naturally aged samples. Oxygen steadily and constantly affects the symmetry of the polypeptide, leading to peptide bond oxidation and scissoring. This oxidation reaction appears minimally influenced by the atmosphere in which the silk has aged. However, it remains unclear whether the proportions of oxidation products (such as ketones, aldehydes, or carboxylic groups in α-keto-acids and dicarboxylic groups) differ under the three degradation conditions. Further analysis using mass spectrometry is warranted to investigate the mechanism of oxidation processes depending on the aging conditions.

The hydrolysis estimator $E_{COOH}$ is more susceptible to water exposure than the aging conditions and thus can give a nice insight into the overall degradation condition of the museum silk sample. The most significant changes in its value have been observed in naturally aged historical silk objects, indicating an increase in -COOH groups, likely resulting from a combination of oxidation and dominant hydrolysis effects over the centuries. For long-term artificial aging, the changes in $E_{COOH}$ are less pronounced but still occur even without the introduction of water (under dry open conditions). The aging conditions do influence the trends in this estimator's value, although the possibility of polypeptide crosslinking should also be considered, as has been suggested by other reports [38,39].

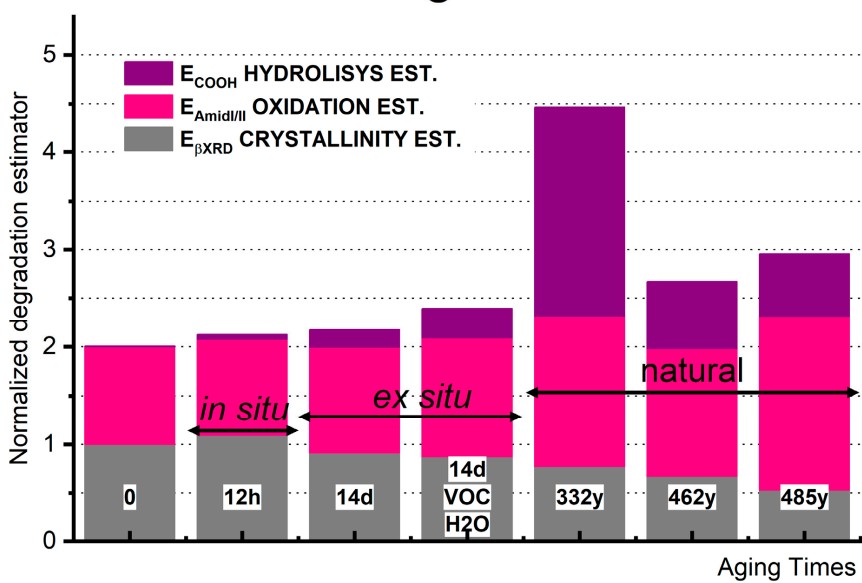

**Figure 10.** Hydrolysis ($E_{COOH}$), oxidation ($E_{AmideI/II}$), crystallinity ($Ec_{FTIR}$), and crystallinity corrected by polarizer ($Ec_{FTIRpol}$) estimators calculated from ATR-FTIR for model silk sample (MS), in situ thermo-aged MS in the air ($isO_2$), ex situ thermo-aged MS in the air ($esO_2$), in the humid closed reactor ($esO_2/VOC/H_2O$) and historic: Inscription Banner (IB); Court Banner (CB); Stanisław Barzi's Funeral Banner (FB).

## 5. Conclusions

Fourier-transformed infrared spectroscopy has been used for over a decade to monitor the secondary structure of silk fibroin-based materials [43]. It was essential to examine this technique and improve the significance of its results.

The oxidation estimator $E_{AmideI/II}$ appears to be the most stable and least affected by the degradation conditions, thus being the most age-correlated estimator. Its consistent increase indicates that peptide bond oxidation occurs from the initial hours of artificial aging and continues throughout long-term degradation under natural and artificial conditions. The oxidation rate is slightly accelerated with the introduction of water, but it is not affected by changes in crystallinity. Fibroin oxidases lead to the formation of ketones (including o-quinone), aldehydes, $\alpha$-keto-acids with carboxylic groups, and dicarboxylic groups along the amino acid chain.

On the other hand, the hydrolysis estimator $E_{COOH}$ is more sensitive to the environment, and its trends change with the introduction of water, as expected. The presence of water accelerates the breaking of amino acid chains and leads to the formation of COOH groups. This hydrolysis strongly impacts the crystalline structure. Water entering in between pleats (as calculated by Cheng, Y. et al. [44]) plays a role in L to D amino acids transition in both parallel and antiparallel $\beta$-sheet domains [41,42]. Still, it cannot be excluded that hydrolysis reaction in dry conditions may not be followed by crosslinking or further oxidation of -COOH groups (as the vibration of free dicarboxylic amino acids (COOH-NH-...-COOH) weakens considerably with prolonged ex situ artificial aging).

Crystallinity for samples with a dichroic ratio *R*-value close to zero exhibits high hydrolysis and oxidation product levels. Polarization of the infrared laser beam ($E_{CPOL}$) enables the determination of lost antiparallel sheet crystalline structures, thereby defining the point at which the FTIR-derived information about crystallinity no longer reflects the sample's overall crystallinity level (*R* below zero). The loss of order in the amino acid structure produces similar products in both antiparallel and parallel pleated domains. This can be explained by the natural conversion of L-amino acids to their D form, altering the hydrogen bonding symmetry in the silk's crystalline regions [41,44].

**Author Contributions:** Conceptualization, J.P.-P. and M.A.K.; methodology, M.A.K.; software, M.A.K.; validation, M.A.K. and J.B.; formal analysis, M.A.K., K.G., M.M.Z.-O., D.P. and M.S.; investigation, M.A.K., J.B. and J.P.-P.; resources, M.A.K. and E.B.; data curation, M.A.K.; writing—original draft preparation, M.A.K.; writing—review and editing, J.B. and M.A.K.; visualisation, M.A.K.; supervision, J.P.-P. and E.B.; project administration, E.B.; funding acquisition, M.A.K. All authors have read and agreed to the published version of the manuscript.

**Funding:** This research was funded by the National Centre of Science (Narodowe Centrum Nauki), grant number NCN FUGA 2016/20/S/ST4/00149.

**Institutional Review Board Statement:** Not applicable.

**Informed Consent Statement:** Not applicable.

**Data Availability Statement:** Data is available online: https://koperska.info/fuga/ (accessed on 29 June 2023).

**Acknowledgments:** The authors thank the Wawel Castle Museum for their help and support.

**Conflicts of Interest:** The authors declare no conflict of interest.

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
