# Peer review of "Ex Situ and In Situ Artificial Thermo-Aging Study of the Natural Degradation of Bombyx mori Silk Fibroin"

_applsci, doi:10.3390/app13169427_

Round 1
Reviewer 1 Report
This article explores the degradation mechanism of silk fibroin using Fourier transform infrared spectroscopy. It is found that the degradation of silk fibroin is influenced by peptide bond oxidation, environmental moisture, and other factors, leading to changes in crystallinity. The presence of water accelerates the oxidation process, and the change in crystallinity is mainly driven by oxidation. Water plays a crucial role in accelerating hydrolysis and causing structural changes. This study provides some information for preserving historical silk artifacts. Before the article is accepted, there is another question that is not very clear. I hope the author can make a supplement or answer.
In addition to the changes of some FITR parameters mentioned by the author, the most direct evidence for the degradation of Fibroin is the degradation of silk fibroin heavy chain protein molecules, which degrades from a large protein to a smaller protein. In my impression, although silk fibroin protein undergoes a certain degree of degradation after degumming at different concentrations, there is no significant change in FITR related parameters. During the artificial aging experiment conducted by the author, if the molecular degradation of silk fibroin can be analyzed in correspondence with the corresponding FITR parameters, this result will more intuitively reveal the degree of silk degradation and find the most effective degradation characteristic peak. If possible, it is recommended that the author supplement this experiment.
Other minor issues,
1. Figure 1B in line 303, please verify.
2. 3.1.3. subheading is over, it means 3.2.4. Since the results section of the article only includes 4 points, it is recommended to change to 3.1 to 3.4, without further grading.
3. In the discussion section, the writing issues such as section 4.1 followed by section 3.1 which indicated that the author is some careless.
4. the conclusion section should be simplify.
Author Response
Dear reviewer,
We extend our sincere thanks for your review of our work. Your feedback is greatly appreciated and will undoubtedly contribute to the refinement and improvement of our research. Your insights are invaluable, and we are grateful for your time and expertise in evaluating our work. Thank you for your thoughtful contribution. Below please find more detailed answers to your questions and remarks.
The changes after the degumming of silk are not investigated here due to the fact that most of the historical silks have already been degummed in the past, and the issue is not of great importance to the current museum conservators. Regarding degummed silk, the characteristic peaks (or, more precisely, peak ratios) were carefully chosen in our previous work. Their ratios already existing in the literature and new ones proposed by our group were verified by numerous analytical techniques to gain insight into the real meaning of the changes observed in the spectra. Observed changes in the peak ratios correspond to the ones observed in the literature.
Ad 1.“Figure 1B” has been corrected for “Figure 3B”
Ad. 2. The numbering has been corrected.
Ad. 3. The numbering has been corrected.
Ad 4 The conclusions have been simplified.
We hope that you approve of our corrections. If you have any further questions or comments, please do not hesitate to provide us with more feedback.
the publication team
Reviewer 2 Report
The title of the manuscript is remarkable. English language has good quality. Figures need some changes. Main text need some modifications. There are some modifications that need to be exerted in the citations.
1. Line 20 is unnecessary. Please delete it.
2. Why Line 30-39 does not have proper reference(s)?
3. All multipple and middle sentence references in all over the manuscript should be reformed
4. Please insert a high resolution version of Figure 1 so that the details of this figure become obviously visible (specially the text of figures).
Note: Figures should have the resolution of 300dpi.
5. Why line 61-65 does not have suitable reference(s)?
6. Line 102- 114 should be mentioned at the end of the section "Introduction". Please transfere these two paragraph at the end of this part and then, rewrite the section based on this change.
7. Why the authors have written the Line 115-128? Please explain why you have written about your prior work in this section? Eas it not better to explain about it in the section "Discussion:? If not, please explain why?
8. Line 133-142 belongs to the section "material and methods" please omit this part.
9. About the section "Discussion"
Please categorize your results based on their
importance from the most important one to the least. After that, discuss about each one of them one by one.
10. About all figures of the manuscript
Please increase the quality (resolution) of figures (the resolution should be 300 dpi) so that the details of each figure (specially texts) become obviously visible
11. Please rewrite the part "Conclusion"
This part should be brief and contain the conclusion of your work based on previous findings. Any other information is unnecessary. (Please conclude your work briefly)
12. Please check and adjust the "Reference list" based on the regulations of reference list of journal. (Titles, doi, the name of journal and ... )
Author Response
Dear reviewer,
We extend our sincere thanks for your review of our work. Your feedback is greatly appreciated and will undoubtedly contribute to the refinement and improvement of our research. Your insights are invaluable, and we are grateful for your time and expertise in evaluating our work. Thank you for your thoughtful contribution. Below please find more detailed answers to your questions and remarks.
Ad.1. It has been deleted.
Ad. 2. References have been added.
Ad. 3 This has been corrected.
Ad 4. The figure resolution has been improved. This might not be visible in PDF, but it is in the Word editable version of the publication.
Ad 5. References have been added.
Ad 6. This has been transferred and corrected.
Ad 7. We believe it is crucial to emphasize, at the outset of this publication, the extensive effort invested in validating the FTIR-derived estimators. By doing so, we aim to comprehensively understand the methodology and its origins in research and results. The inquiries raised from the prior work conducted by our group have directly influenced this current study, making it essential to acknowledge and acknowledge it at the beginning.
Ad 8. This line was omitted
Ad 9. Thank you for this remark. This has been taken care of.
Ad.10. This has been corrected.
Ad 11. Conclusions have been rewritten.
Ad.12 The reference list has been corrected.
We hope that you approve of our corrections. If you have any further questions or comments, please do not hesitate to provide us with more feedback.
the publication team
Round 2
Reviewer 2 Report
I checked and did not find any point for correction. Good luck